# Genome-Wide Insight into Profound Effect of Carbon Catabolite Repressor (Cre1) on the Insect-Pathogenic Lifecycle of *Beauveria*
*bassiana*

**DOI:** 10.3390/jof7110895

**Published:** 2021-10-23

**Authors:** Rehab Abdelmonem Mohamed, Kang Ren, Ya-Ni Mou, Sheng-Hua Ying, Ming-Guang Feng

**Affiliations:** MOE Laboratory of Biosystems Homeostasis & Protection, College of Life Sciences, Zhejiang University, Hangzhou 310058, China; 11807145@zju.edu.cn (R.A.M.); 11807120@zju.edu.cn (K.R.); 11907036@zju.edu.cn (Y.-N.M.); yingsh@zju.edu.cn (S.-H.Y.)

**Keywords:** entomopathogenic fungi, carbon catabolite repression, gene expression and regulation, host infection, virulence, biological control

## Abstract

Carbon catabolite repression (CCR) is critical for the preferential utilization of glucose derived from environmental carbon sources and regulated by carbon catabolite repressor A (Cre1/CreA) in filamentous fungi. However, a role of Cre1-mediated CCR in insect-pathogenic fungal utilization of host nutrients during normal cuticle infection (NCI) and hemocoel colonization remains explored insufficiently. Here, we report an indispensability of Cre1 for *Beauveria*
*bassiana*’s utilization of nutrients in insect integument and hemocoel. Deletion of *cre1* resulted in severe defects in radial growth on various media, hypersensitivity to oxidative stress, abolished pathogenicity via NCI or intrahemocoel injection (cuticle-bypassing infection) but no change in conidial hydrophobicity and adherence to insect cuticle. Markedly reduced biomass accumulation in the Δ*cre1* cultures was directly causative of severe defect in aerial conidiation and reduced secretion of various cuticle-degrading enzymes. The majority (1117) of 1881 dysregulated genes identified from the Δ*cre1* versus wild-type cultures were significantly downregulated, leading to substantial repression of many enriched function terms and pathways, particularly those involved in carbon and nitrogen metabolisms, cuticle degradation, antioxidant response, cellular transport and homeostasis, and direct/indirect gene mediation. These findings offer a novel insight into profound effect of Cre1 on the insect-pathogenic lifestyle of *B. bassiana*.

## 1. Introduction

Fungal growth and development rely upon the utilization of nutrients in the forms of environmental carbon and nitrogen sources. Carbon catabolite repression (CCR) is a canonical mechanism underlying the preferential utilization of glucose derived from a mixture of carbon sources. In budding yeast, CCR is regulated by three transcription factors (Mig1–3), which feature DNA-binding Cys_2_His_2_ zinc-finger motifs and mediate the expression of glucose-repressed genes vectoring Mig-binding sites in their promoter regions [1,2,3]. The activity of Mig1 is linked to nuclear localization and protein modification, such as its phosphorylation to alleviate the repressive activity [4]. The nucleocytoplasmic shuttling of Mig1 is controlled by AMP-activated protein kinase Snf1p/SnfA [5,6], and also by hexose kinase Hxk2 in the presence of glucose and mannose and by Hxk1 or Hxk2 in the presence of fructose [7]. In fission yeast, the Mig1 orthologue Scr1 mediates many genes involved in carbon metabolism, hexose uptake, gluconeogenesis and tricarboxylic acid (TCA) cycle [8].

In filamentous fungi, carbon catabolite repressor A (CreA/Cre1) orthologous to the yeast Mig1 is well known to mediate CCR in *Aspergillus nidulans*. The loss-of-function mutation of *creA* led to reduced growth and conidiation in glucose-inclusive media and depressed pathways required for lactose and starch utilization [9,10]. Indeed, CreA acts as unique DNA-binding protein essential for the repression of alcohol dehydrogenase gene *alcA* [11], and is regulated at transcriptional and posttranscriptional levels [12]. The CreA activity acts as part of a more complex regulatory circuit, as revealed by microarray analysis of *creA* mutants grown on glucose or ethanol [13], but is not necessarily linked to transcriptional autoregulation, cellular translocation and/or degradation [14]. More recent studies have uncovered diverse routes of CreA regulation in *A. nidulans*. CreA depends partially on de novo protein synthesis and is regulated in part by ubiquitination [15].The F-Box protein Fbx23 serves as part of an SCF (Skp-Cullin-F-box) ubiquitin ligase complex bridged to the CreA-SsnF-RcoA repressor complex, leading to degradation of the complex under derepressing conditions, disassociation of CreA from the complex and its import into the nucleus in the presence of glucose [16]. The direct phosphorylation of CreA S262 by casein kinase A (CkiA) [17] and the indirect phosphorylation of CreA S319 by cyclic AMP (cAMP)-dependent protein kinase (PkaA) in the presence of glucose suggest a requirement of CreA phosphorylation for the fungal CCR [18].Three other phosphosites (S262, S268, and T308) have also proved important for CreA protein accumulation, subcellular localization, DNA binding, and repressed enzyme activities in *A. nidulans* [19]. In protein-protein interaction studies, the mitogen-activated protein kinase (MAPK) MpkB (Fus3) and the MAPK kinases Ste7 and PbsA (Pbs2) were shown to serve as part of a protein complex that regulates subcellular localization of CreA in the presence of xylan and is dissociated for the performance of CCR upon the addition of glucose [20].

Aside from the intensive studies in the model fungi, CreA- or Cre1-mediated CCR also has been studied in other filamentous fungi, including mycoparasitic *Trichoderma* and plant pathogens. In *Trichoderma reesei*, Cre1 may regulate transcriptional expression of both the xylanase gene *xylI* and the cellobiohydrolase gene *cbh1* [21,22], and its DNA- binding activity relies upon site-specific S241 phosphorylation [23]. Transcriptional profiling of *T. reesei* wild-type and Δ*cre1* strains grown on a glucose-inclusive medium revealed a role for Cre1 in mediating expression levels of ~250 genes, many of which encode transporters and functionally unknown proteins [24]. Nucleocytoplasmic shuttling of Cre1 is evidently important for the mediation of CCR by Cre1 imported from the cytoplasmic pool [25]. Intriguingly, the functions of *cre1* orthologues vary largely in different fungal lineages. The *cre1* gene from *Sclerotinia*
*sclerotiorum* was shown to complement the *A. nidulans*
*creA* mutation repressing the *alcA* expression but to not complement the *mig* deficiencies in *S. cerevisiae* [26]. Deletion of *creA*in *Alternaria brassicicola* had no impact on the fungal growth and virulence [27], contrasting with an inability to disrupt *cre1* in a *Fusarium oxysporum* strain [28]. CCR disruption of *Penicillium funiculosum* by site- specific mutation of Mig1 resulted in facilitated hyphal growth and branching, enhanced carbon source usage, and increased cellulase activity [29]. In contrast, deletion of *creA* led to reduced radial growth and stouter hyphae in the thermophilic fungus *Humicolainsolens* [30].

The Cre1-mediated CCR is of special interest for insect-pathogenic fungi, which utilize scant nutrients in host integument during hyphal invasion into host body, serve as sources of environment-friendly insecticides and are represented best by *Beauveria*
*bassiana* and the *Metarhizium*
*anisopliae* complex [31,32,33]. Previously, a role of CRR1 (homologous to Cre1) was noted in the carbon regulation of the cuticle-degrading protease Pr1 in *M. anisopliae* [34]. Antisense knockdown expression of *creA* (*BbcreA*) in *B. bassiana* led to derepressed Pr1 activity, unaffected virulence, and minor growth defects in the presence of ethanol or xylose [35]. Indeed, the Pr1 family comprises 11 proteases, which are collectively important for normal infection via cuticular penetration but individually contribute limitedly or little to the virulence of *B. bassiana* [36]. The pleiotropic effect of *creA* in *B. bassiana* was subsequently elucidated with phenotypes of knockout and complementation mutants [37]. In the study, deletion of *BbcreA* resulted in severe growth defects on rich and minimal media, reduced conidiation capacity, derepressed cuticle- degrading enzymes, attenuated virulence, and an interesting ‘cell-lytic phenotype’, which was observed at the base of germ tubes on various media and most worsened on a minimal medium containing peptone as sole nitrogen source. Recently, the histone lysine methyltransferase KMT2/SET1 was found to catalyze the trimethylation of histone H3K4 as an epigenetic mark that activates the *cre1* expression leading to transcriptional upregulation of *hyd4*, a critical hydrophobin gene essential for approssorial formation and pathogenicity of *Metarhizium robertsii*, unveiling that the KMT2-Cre1-Hyd4 pathway mediates the fungal pathogenesis [38]. In the study, deletion of *cre1* in *M. robertsii* resulted in a similar virulence reduction (~50%) as seen in the Δ*BbcreA* mutant [37]. In *B. bassiana*, classes I and II hydrophobins (Hyd1 and Hyd2), which are required for hydrophobin biosynthesis and assembly into an outermost rodlet-bundle layer of conidial coat determinant to conidial hydrophobicity and adherence to insect cuticle [39], are both homologous to Hyd4 in *M. robertsii*, leading to discovery of the SET1-Cre1-Hyd1/2 pathway that mediates not only fungal pathogenicity but also asexual cycle in vitro [40]. The previous studies demonstrate a crucial role for Cre1 in mediating the expression of critical hydrophobin gene(s) required for appressorial formation and pathogenesis in *M. roberstii* or for conidial hydrophobicity/adherence and insect pathogenicity in *B. bassiana*, which does not form typical approssoria during hyphal invasion into insect body [41]. The previous studies also reveal that derepressed enzymes, including proteases, chitinases, esterases and hydrocarbon-assimilating enzymes targeting various components of insect cuticle [42,43], concurred with attenuated virulence when *cre1* was downregulated [35] or deleted [37]. So conflicting phenotypes suggest that the regulatory role for Cre1 in fungal insect-pathogenic lifecycle remains understood insufficiently and hence to be explored for deeper insight into its pleiotropic effect on the fungal lifecycle. This study aims to reevaluate the role of *cre1* in the lifecycle in vivo and in vitro of *B. bassiana* by multi-phenotypic analyses of its deletion and complementation mutants and gain a genome-wide insight into global effect of *cre1* through transcriptomic analysis of the deletion mutant versus the parental wild-type strain. Our data demonstrate an indispensability of *cre1* for fungal host infection and hemocoel colonization and its crucial role in the direct/indirect mediation of 1881 genes that are involved in a large array of cellular processes and events required for the lifecycle in vivo and in vitro of *B. bassiana*.

## 2. Materials and Methods

### 2.1. Subcellular Localization of Cre1

The plasmid pAN52-C-gfp-bar (C: the cassette 5′-*Pme*I-*Spe*I-*Eco*RV-*Eco*RI-*Bam*HI-3′ under the control of the homologous promoter P*tef1* [44,45]) was used as backbone to generate transgenic strains expressing the fusion protein Cre1-GFP in the wild-type strain *B. bassiana* ARSEF 2860 (designated WT herein), which is a typical strain used in genome sequencing and annotation [46] and features faster growth, greater conidiation capacity, higher virulence and broader host spectrum than some other research strains, such as the strain ATCC 90517 used for generation of *BbcreA* null mutants [37]. Briefly, the coding sequence of *cre1* (locus tag BBA_05136) was amplified from the WT cDNA with paired primers (Appendix A) and ligated to the N-terminus of *gfp* (green fluorescence protein gene) in the plasmid linearized with *Bam*HI/*Xma*I, followed by integrating pAN52-*cre1*-gfp-bar into WT via *Agrobacterium*-mediated transformation and screening of putative transformants by *bar* resistance to phosphinothricin (200 μg/mL). A transgenic strain showing strong green signal was incubated on SDAY [Sabouraud dextrose agar (4% glucose, 1% peptone and 1.5% agar) plus 1% yeast extract] for conidiation at an optimal regime. The resultant conidia were incubated at 25 °C for 48 h in SDBY (i.e., agar-free SDAY) on a shaking bed (150 rpm). The resultant culture was rinsed repeatedly in sterile water and resuspended in minimal Czapek-Dox broth (CDB: 3% sucrose, 0.3% NaNO_3_, 0.1% K_2_HPO_4_, 0.05% KCl, 0.05% MgSO_4_ and 0.001% FeSO_4_) amended with 3% xylose as sole carbon source, followed by a 3 h shaking incubation. Hyphal samples taken from the SDBY culture and that triggered with xylose were stained with the nuclear dye DAPI (4′,6′-diamidine-2′-phenylindole dihydrochloride; Sigma-Aldrich, Shanghai, China), followed by laser scanning confocal microscopic (LSCM) analysis to determine subcellular localization of Cre1-GFP in response to glucose or xylose.

### 2.2. Generation of Cre1 Mutants

An intron-free full-length coding sequence (1158 bp) of *cre1* with flanking regions (524 bp) was completely deleted from the WT strain by homologous recombination of its 5′ and 3′ flanking fragments (1520 and 1521 bp respectively) separated by the *bar* marker (Appendix A) and complemented into an identified Δ*cre1* mutant by ectopic integration of its full-length coding sequence via the aforementioned transformation. Briefly, the two fragments were amplified from the WT DNA and ligated to the enzyme sites of *Eco*RI/ *Hin*dIII and *Xba*I/*Hpa*I in p0380-bar, forming p0380-5′F-bar-3′F. The full-length coding sequence of *cre1* with flank regions (3639 bp in total) was amplified from the WT DNA and ligated to the sites of *Hin*dIII/*Xba*I in p0380-sur-gateway to exchange for the gateway fragment, yielding p0380-sur-*cre1*. The constructed plasmids were transformed into the WT and Δ*cre1* strains, respectively, as aforementioned. Putative mutant colonies were screened by the *bar* resistance to phosphinothricin (200 μg/mL) or the *sur* resistance to chlorimuron ethyl (10 μg/mL), followed by verification of expected recombinant events via PCR (Appendix A) and real-time quantitative PCR (qPCR). The deletion mutant (DM) Δ*cre1* showing abolished *cre1* expression and the complementation mutant (DM) Δ*cre1::cre1* with *cre1* expression restored to the WT level (Appendix A) were evaluated in parallel with the WT strain in the following experiments including three independent replicates unless specified otherwise. All primers used for targeted gene manipulation and detection are listed in Appendix A.

### 2.3. Examination of Cell-Lytic Phenotype

The previous Δ*BbcreA* mutant created with the strain ATCC 90517 displayed the most conspicuous phenotype of cell lysis on CDA (i.e., CDB plus agar) amended with 0.5% peptone as sole nitrogen source [37]. To verify whether this interesting phenotype occurred in the absence of *cre1*, conidia of the mutant and WT strains were spread on the cellophane-overlaid plates of peptone-amended CDA and incubated for 24 h at optimal 25 °C. Culture samples were taken at the end of 12- or 24-h incubation, stained with the cell wall-specific dye calcofluor white and examined through LSCM analysis. In addition, ultrathin sections of hyphal cells were examined with transmission electron microscopy (TEM) to show a status of cell wall integrity in the presence or absence of *cre1*.

### 2.4. Assays for Virulence and Virulence-Related Cellular Events

The third-instar larvae of greater wax moth (*Galleria*
*mellonella*) were used to assay the virulence of each strain in two infection modes. Briefly, three groups (replicates) of ~35 larvae per strain were separately inoculated by topical application (immersed for 10 s in 40 mL aliquots) of a 10^7^ conidia/mL suspension for normal cuticle infection (NCI). Alternatively, 5 μL aliquot of a 10^5^ conidia/mL suspension was injected into the hemocoel of each larva (i.e., ~500 conidia injected per larva) in each group for cuticle-bypassing infection (CBI). Three groups of larvae immersed in or injected with sterile water containing 0.02% Tween 80 (vector of conidial suspension) were used as controls for the corresponding infection mode. All treated groups were maintained at 25 °C for up to 15 days and examined daily for survival/mortality records. Median lethal time (LT_50_) was estimated as an index of virulence from the time-mortality trend of each group by modeling analysis.

Some cellular events crucial for hyphal invasion into insect body and subsequent hemocoel colonization were examined for in-depth insight into possible virulence differences between DM and control (WT and CM) strains. First, conidial adherence to insect cuticle, a critical trait for NCI initiation, was assayed on locust hind wings as described previously [41]. Briefly, 5 μL aliquots of a 10^7^ conidia/mL suspension in sterile water containing no surfactant were spotted on the central areas of hind wings attached to 0.7% water agar. After an 8 h incubation at 25 °C. Counts of conidia were made immediately from three microscopic fields of each wing and made again after 30 s washing of less adhesive conidia in sterile water. Percent ratios of pre-wash versus post-wash counts were computed as conidial adherence of each strain to wing cuticle with respect to the WT standard. Second, total activities (U/mL) of extracellular (proteolytic, chitinolytic and lipolytic) enzymes (ECEs) and Pr1 proteases required for cuticular penetration [36,42] were assayed from the supernatants of 3-day-old liquid cultures grown in 50 mL aliquots (10^6^ conidia/mL) of CDB amended with 0.3% bovine serum albumin (BSA) as sole nitrogen source and enzyme inducer, as described previously [36,47]. Biomass levels were also assessed from the CDB-BSA cultures. In addition, hemolymph samples were taken from the larvae surviving 72 or 288 h post-CBI and examined under a microscope to reveal the presence/absence and abundance of hyphal bodies (i.e., blastospores), which usually form from the hyphae arrived in the host hemocoel, propagate rapidly by yeast-like budding to accelerate host death from mummification and hence reflect the status of fungal hemocoel colonization linked tightly to virulence [48,49].

### 2.5. Assays for Radial Growth, Stress Response, Conidiation Capacity and Conidial Quality

Hyphal invasion into insect body after conidial germination on insect integument relies upon hyphal growth under normal and stressful conditions. Thus, fungal colonies were initiated by spotting 1 μL aliquots of a 10^6^ conidia/mL suspension on the plates of rich SDAY, 1/4 SDAY (amended with 1/4 of each SDAY nutrient), minimal CDA and CDAs amended with different carbon (glucose, trehalose, maltose, lactose, galactose, fructose, glycerol and sodium acetate) or nitrogen (NH_4_Cl, NaNO_2_ and NH_4_NO_3_) sources. The spotting method was also used to initiate colony growth on the plates of 1/4 SDAY alone (control) or supplemented with NaCl (0.8 M) or sorbitol (1.0 M) for osmotic stress, menadione (0.04 mM) or H_2_O_2_ (4 mM) for oxidative stress, and Congo red (10 μg/mL) or calcofluor white (10 μg/mL) for cell wall stress, respectively. After a 6-day incubation at 25 °C, the diameter of each colony was estimated as a growth index with two measurements taken perpendicular to each other across the center. Relative growth inhibition (RGI) for the response of each strain to a given stress was assessed as (*d*_c_ − *d*_s_)/*d*_c_ × 100, where *d*_c_ and *d*_s_ denote the diameters of control and stressed colonies respectively. In addition, total activities (U/mg) of superoxide dismutases (SODs) and catalases were assayed from the protein extracts of 3-day-old SDAY cultures with SOD Activity Assay Kit (Sigma-Aldrich) and Catalase Activity Assays Kit (Jiancheng Biotech, Nanjing, China) following the manufacturers’ guides, respectively.

Cultures used for assessments of conidiation capacity and biomass accumulation were initiated by spreading 100 μL aliquots of a 10^7^ conidia/mL suspension on SDAY plates (9 cm diameter) overlaid with or without cellophane and incubated for 8 days at the optimal regime of 25 °C in a light/dark (L:D) cycle of 12:12 (h). During the period of incubation, samples were taken from 3-day-old cultures and stained with calcofluor white, followed by microscopic examination of conidiation status. From day 4 onwards, three samples were taken daily from each plate culture with a cork borer (5 mm diameter). Conidia on each sample were released into 1 mL of aqueous 0.02% Tween 80 by 10 min supersonic vibration. The conidial yield was estimated by assessing the concentration of the conidial suspension with a hemocytometer and converting the concentration to the number of conidia per unit area (cm^2^). The cellophane-overlaid SDAY cultures were collected for assessments of biomass levels every 2 days from day 3 onwards.

For each strain, conidial hydrophobicity crucial for conidial adherence to insect cuticle [41] was assayed in a diphasic (aqueous-organic) system, followed by scanning electron microscopic (SEM) examination for the presence/absence of hydrophobin rodlet bundles in the outermost layer of conidial coat, as described previously [45,50,51]. To assess the antioxidant capability of conidia important for response to superoxide reactive species (ROS) generated by host immunity defense during NCI or CBI [52], 100 μL aliquots of a 10^6^ conidia/mL suspension were spread evenly onto the plates of trehalose-peptone agar (TPA, i.e., CDA amended with 3% trehalose and 0.3% peptone as carbon and nitrogen sources respectively) alone (control) or supplemented with gradient concentrations of menadione (0.01–0.04 mM) and H_2_O_2_ (1–4 mM), respectively, followed by an incubation at 25 °C. From 4 h incubation onwards, germinated and non-germinated conidia were counted hourly from three microscopic fields of each plate until no more germination change. The time-germination trends fit the equation *R_g_* = 1/[1 + exp(*a* + *rt*], where *t* and *R_g_* denote incubation time (h) and germination ratio (≤1) while *a* and *r* are parameters to be fitted. Median germination time (GT_50_ = −*a*/*r*) was estimated as an index of conidial sensitivity to a given concentration of either oxidant from each of the fitted trends.

### 2.6. Transcriptomic Analysis

Construction and analysis of *cre1*-specific transcriptome were performed by preparing three 3-day-old cultures (replicates) of the Δ*cre1* and WT strains incubated on cellophane-overlaid SDAY plates as aforementioned and sending them to Lianchuan BioTech Co. (Hangzou, China) for services. Total RNA extraction from each culture, isolation of mRNA from total RNA, fragmentation of mRNA, and syntheses of first- and second-strand cDNAs were performed as described previously [53]. Purified double-stranded cDNA was repaired by adding a single adenine to its end. The final cDNA library was sequenced on an Illumina Novaseq 6000 platform.

All clean tags were generated by filtration of raw reads from the sequencing and mapped to *B. bassiana* genome [46]. All data were normalized as fragments per kilobase of exon per million fragments mapped. Differentially expressed genes (DEGs) were identified at significant levels of both log_2_ *R* ≤ −1 (downregulated) or ≥1 (upregulated) and *q* < 0.05, annotated with gene information in NCBI databases, and subjected to Gene Ontology (GO) analysis (http://www.geneontology.org/, accessed on 19 October 2021) for enrichments of GO terms to three function classes (*p* < 0.05) and Kyoto Encyclopedia of Genes and Genomes (KEGG) analysis (http://www.genome.jp/kegg/, accessed on 19 October 2021) for pathway enrichment (*p* < 0.05).

### 2.7. Real-Time Quantitative PCR (qPCR) Analysis

Transcript levels of *cre1* and 27 genes were assessed via qPCR to gain an insight into altered phenotypes and/or verify a validity of transcriptomic data following prvevious protocol [53]. Briefly, the 3-day-old SDAY cultures of the DM and control strains were prepared as aforementioned. Total RNAs were extracted from the cultures under the action of RNAiso Plus Kit (TaKaRa, Dalian, China), and reversely transcribed into cDNAs under the action of Prime Script RT reagent kit (TaKaRa), respectively. The cDNA samples derived from three independent cultures of each strain were used as templates in the qPCR analysis to assess transcripts of all concerned genes with paired primers (Appendix A) using SYBR Premix *ExTaq* (TaKaRa). Aside from *cre1*, the examined genes included those involved in hydrophobicity (*hyd1*–*5*), degradation of superoxide anions (*sod1**–5*) and H_2_O_2_ (*cat1*–*6*), conidiation and conidial maturation (*brlA*, *abaA*, *wetA*, *vosA*, *frq1* and *vvd*), cell wall composition and signaling (*CWP* and *mkk1*), and transcriptional regulation (*cfp*, *TF1* and *TF2*). The fungal β-actin gene was used as a reference. A threshold-cycle (2^−ΔΔCt^) method was used to compute relative transcript levels of each genein the mutant cultures with respect to the WT standard.

### 2.8. Statistical Analysis

All data from the experiments of three independent replicates were subjected to one-way analysis of variance, followed by Tukey’s honestly significant difference (HSD) test for phenotypic differences among the tested strains.

## 3. Results

### 3.1. Subcellular Localization of Cre1

Bioinformatic information of Cre1 (NCBI accession code: EJP65725; 385 amino acids) in this study is identical to BbCreA investigated previously [37]. As revealed with LSCM images, the green fluorescence-tagged Cre1-GFP fusion protein accumulated in both cytoplasm and nuclei of hyphal cells from the culture grown in SDBY containing glucose, and overlapped weakly with the stained color (shown in red) in small nuclei (left panels in Figure 1A). In contrast, a 3-h exposure to xylose as sole carbon source in amended CDB resulted in much heavier accumulation of the fusion protein in the nuclei than in the cytoplasm, as indicated by bright yellow formed by overlapped green and red images (right panels in Figure 1A). Also as a result of the exposure, the nuclei became much larger in size and irregular in shape. These images demonstrated much more import of Cre1 into the nuclei of *B. bassiana* from the cytoplasm in the presence of xylose than of glucose and were distinctive from Cre1 equally localized to the *T. reesei* nuclei under cellulase inducing (lactose) and repressing (glucose) conditions [25] and also from CreA imported into the *A. nidulans* nuclei in the presence of glucose [16], suggesting a variation in Cre1- or CreA-mediated CCR among filamentous fungi.

### 3.2. No Cell-Lytic Phenotype Occurs in the Absence of Cre1

Cell lysis at the base of germ tubes was revealed as a major phenotype in the previous Δ*BbcreA* mutant and most worsened on CDA amended with peptone as sole nitrogen source [37]. To verify this phenotype, LSCM analysis of germ tubes formed after a 12 or 24 h incubation of conidia on the mentioned medium were carried out. Surprisingly, our Δ*cre1* mutant showed no cell-lytic phenotype at all regardless of the germ tubes observed at the end of 12 or 24 h incubation (Figure 1B). For the mutant, all cell walls of conidia, germ tubes and extending hyphae were well defined by the cell wall staining, leading to no sign of cell wall lysis at the base of each germ tube. TEM images for the ultrathin sections of hyphal cells also revealed little difference in hyphal cell wall integrity between the mutant and the control strains (Figure 1C).

### 3.3. Indispensability of Cre1 for NCI and Hemocoel Colonization

In the standardized bioassays, the control strains killed all *G.*
*mellonella* larvae within 11 days post-NCI or 8 days post-CBI, contrasting with very few died from mycosis caused by the Δ*cre1* mutant 15 days post-NCI or 10 days post-CBI (Figure 2A). Consequently, LT_50_s for the control strains against the model insect on average were 6.3 (±0.19) days via NCI and 4.3 (±0.12) days via CBI, but no LT_50_ was computable for the Δ*cre1* mutant in either infection mode. Its inability to cause substantial insect mortality via NCI or CBI was further verified in two more repeated bioassays (three replicates per capita; data not shown).

For insight into an inability for the Δ*cre1* mutant to initiate NCI, conidial adherence essential for NCI initiation was assayed on locust hind wings. As a result, no variability in conidial adherence to the wing cuticle (*F*_2,6_ = 0.34, *p* = 0.72) was found among the Δ*cre1* and control strains (Figure 2B). In the supernatants of 3-day-old CDB-BSA cultures, total activities of secreted ECEs and Pr1 proteases required for successful NCI were reduced by 96% and 94% in Δ*cre1* relative to the WT strain (left panel in Figure 2C). The two reductions diminished to 71% and 55% respectively when the effect of decreased biomass accumulation (right panel in Figure 2C) was deducted by estimating the activities of those enzymes secreted per milligram of biomass. Microscopic examination revealed the presence of abundant hyphal bodies formed by the control strains and of lysing host hemocytes in the hemolymph samples taken at 72 h post-injection (upper panels in Figure 2D). In contrast, the samples from the larvae surviving the injection with the Δ*cre1* conidia showed an aggregation of many more intact hemocytes, but no hyphal bodies, at the time of 72 or even 288 h post-CBI (lower panels in Figure 2D). Consequently, almost all of the larvae lived very well at the end of 12 days after injection with the mutant conidia while cadavers of the larvae killed by the control strains via CBI were covered with heavy fungal outgrowths conidiating abundantly at the same time (Figure 2E).

Both NCI and CBI abolished in the absence of *cre1* indicated an indispensability of *cre1* for *B. bassiana*’s insect pathogenicity and hemocoel colonization, which is crucial for proliferation in vivo by yeast-like budding to accelerate host death from mummification. The abolished NCI was partially associated with reduced secretion of cuticle-degrading enzymes but not associated with any change in conidial adherence to insect cuticle, suggesting involvements of some other cellular events in the infection course of the Δ*cre1* mutant.

### 3.4. Essential Role of Cre1 in Radial Growth and Antioxidant Response

Deletion of *cre1* resulted in severe defects in radial growth on the plates of rich SDAY, 1/4 SDAY, minimal CDA and CDAs amended with different carbon or nitrogen sources (Figure 3A). Compared to the WT strain, the Δ*cre1* mutant displayed a reduction of radial growth by ~53% on both SDAY and 1/4 SDAY, by 70% on CDA, and by a range from 67% (galactose) to 82% (trehalose) on the amended CDAs under normal culture conditions (Figure 3B).

Due to the mutant’s severe growth defect on CDA, cellular responses to different types of stresses were assayed on 1/4 SDAY at 25 °C. As a result, the Δ*cre1* mutant became hypersensitive to oxidative stress induced by menadione (superoxide anions-generating compound) and also more sensitive to the other oxidant H_2_O_2_ than to the cell wall stress induced by Congo red or calcofluor white (Figure 3C). Based on relative growth inhibition of colony growth under each stress, cellular sensitivities to menadione (0.04 mM) and H_2_O_2_ (4 mM) were elevated by 80% and 35% respectively in Δ*cre1* relative to the WT strain (Figure 3D), contrasting with less than 10% increase in its sensitivity to either cell wall stressor (10 μg/mL). However, all tested strains displayed equal responses to hyperosmotic stress induced by NaCl (0.8 M) or sorbitol (1.0 M). The mutant’s sensitivities to the two oxidants were further assessed during conidial germination. The mutant’s GT_50_ (8.2 h) for 50% of conidial germination at 25°C differed insignificantly (Tukey’s HSD, *p* > 0.05) from the WT estimate (8.0 h) but prolonged markedly by 25%, 98% and 133% in comparison to the WT estimates of 8.9, 8.9 and 10.6 h in the presence of 0.01, 0.02 and 0.03 mM menadione, respectively (left panel in Figure 3E). Notably, inclusion of 0.04 mM menadione in the medium resulted in a WT’s GT_50_ of 12.9 h but no germination at all for Δ*cre1*, giving an explanation for its hypersensitivity to the oxidant during colony growth. Also, the mutant’s GT_50_ was prolonged by 14–44% in comparison to the WT’s estimates of 15.5, 18.3, 25.2 and 25.9 h in the medium containing 1, 2, 3 and 4 mM H_2_O_2_ respectively (right panel in Figure 3E).

Next, total activities of SODs and catalases required for degrading superoxide anions and H_2_O_2_ respectively [52] were measured from the protein extracts isolated from 3-day-old SDAY cultures. Compared to the WT strain, the Δ*cre1* mutant displayed a 14% increase in total SOD activity and a 42% decrease in total catalase activity (Figure 3F). Subsequent qPCR analysis revealed marked repression of two genes (*cat2*/*catB* and *cat5*/*catP*) determinant to the fungal catalase activity [54] and upregulated expression of most SOD-coding genes except insignificant downregulation of *sod2*, which encodes cytoplasmic MnSOD as a dominant contributor to total SOD activity in *B. bassiana* [55]. Obviously, the activities of those antioxidant enzymes correlated with transcriptional changes of their coding genes in the absence of *cre1*.

All changes occurred in the absence of *cre1* were well restored to the WT levels by targeted gene complementation, indicating an essentiality of Cre1 for the redial growth and antioxidant response of *B. bassiana*.

### 3.5. Impact of Cre1 Deletion on Aerial Conidiation and Conidial Hydrophobicity

Conidiation capacity crucial for the fungal survival and dispersal in host habits was quantified from the SDAY cultures initiated by spreading 100 μL aliquots of a 10^7^ conidia/mL suspension and incubated for 8 days at the optimal regime of 25 °C and L:D 12:12. Microscopic examination of samples taken from 3-day-old cultures revealed an initial conidiation status of Δ*cre1* as seen in the control strains at the sampling time (Figure 4A). In the Δ*cre1* cultures, the initial conidiation status correlated with insignificant transcript changes of *brlA* and *abaA*, two key activator genes of central developmental pathway (CDP) indispensable for aerial conidiation [56], while the expression levels of downstream *wetA* and *vosA* required for conidiation and conidial maturation [57] were reduced differentially (Figure 4B). Compared to the WT strain, the mutant showed a marked decrease of conidial yield by 98% on day 4, 84% on day 5, and ~72% on days 6–8, accompanied by a decrease of biomass accumulation by 87% on day 3, 85% on day 5 and 75% on day 7 (Figure 4C). Indeed, the 7- or 8-day-old Δ*cre1* culture was covered by a thin layer of conidial powder containingvery limited hyphal debris. For the Δ*cre1* and control strains, interestingly, conidial yields estimated per milligram of biomass showed insignificant variation on day 5 (*F*_2,6_ = 0.37, *p* = 0.71) or 7 (*F*_2,6_ = 4.41, *p* = 0.07), leading to a mean (±SD) yield of 2.05 (±0.18) × 10^7^ conidia/mg on day 5 and of 2.58 (±0.16) × 10^7^ conidia/mg on day 7. Obviously, the reduced conidial yield was attributed to decreased biomass accumulation rather than to blocked CDP in the absence of *cre1*.

Moreover, the indices of conidial hydrophobicity assessed in the aqueous-organic system showed no variability (*F*_2,6_ = 0.70, *p* = 0.53) among the Δ*cre1* and control strains tested (Figure 4D). Electronic scanning of conidial surfaces revealed seemingly more, but smaller or shorter, hydrophobin rodlet bundles assembled onto the conidial coat of Δ*cre1* in comparison to those assembled onto the coat of each control strain (Figure 4E). Previously, *hyd1* and *hyd2* were characterized as two key genes essential for hydrophobin biosynthesis and assembly into the rodlet bundles of conidial coat in *B. bassiana* [39]. As a result of qPCR analysis, either *hyd1* or *hyd**2* expression was upregulated in Δ*cre1* relative to the WT strain despite differential suppression of three other genes (*hyd3*–*5*) that encode hydrophobin-like proteins but remain functionally unknown yet (Figure 4F). For the mutant, no change in conidial hydrophobicity was consistent with an intact outermost rodlet-bundle layer of conidial coat, which showed a ‘bold’ phenotype when *hyd1* and *hyd2* were both deleted [39] or sharply repressed in the absence of a gene encoding the cysteine-free protein CFP [45], the cosubunit Ssr4 of chromatin-remodeling SWI/SNF and RSC complexes [50] or the fifth subunit Csn5 of COP9 signalosome complex [51]. The subtle change in the morphology of assembled rodlet bundles were likely associated with upregulated expression of *hyd1* and *hyd2* required for hydrophobin biosynthesis and assembly in *B. bassiana*.

### 3.6. Genome-Wide Insight into Pleiotropic Effect of Cre1

The transcriptome constructed by sequencing the cDNAs derived from three independent cultures of the Δ*cre1* and WT strains comprised 10,365 genes, including 1881 DEGs (up/down ratio: 764:1117; the same meaning for all ratios mentioned below) (Figure 5A, Appendix A). The identified DEGs took 18.15% in the whole genome, suggesting a dependence of their expression on Cre1. Intriguingly, 32% of those DEGs encode hypothetical proteins (169:416), which take 25.3% in the annotated *B. bassiana* genome [46], and DUF (domain of unknown function) containing proteins (7:9), suggesting a greater role for Cre1 in transcriptional activation of certain functionally unknown genes. Among the identified DEGs, 35 were repressed as if deleted (log_2_ *R* ≤ −6.98), 54 repressed by 95–99% (−6.6 ≤ log_2_ *R* ≤ −4.34), and 62 repressed by 90–95% (−4.26 ≤ log_2_ *R* ≤ −3.32).

The identified DEGs were enriched to 88 GO terms of three function classes at the significance of *p* < 0.05 (Figure 5B, Appendix A). There were 895 DEGs (354:541) enriched to Cellular Component comprising five GO terms, including mainly cellular component (303:458), integral component of membrane (40:52) and extracellular region (9:28). Enriched to Molecular Function were 44 GO terms and 977 DEGs (375:607). These GO terms were involved mainly in molecular function (200:255), oxidoreductase activity (45:67), catalytic activity (22:38), iron ion binding (12:26), heme binding (22:38), electron carrier activity (6:22), monooxygenase activity (6:13), nucleoside-triphosphatase activity (6:13), flavin adenine dinucleotide binding (7:11), transmembrane transporter activity (7:9), S-adenosylmethionine-dependent methyltransferase activity (7:8), lipase/triglyceride lipase activity (5:9), sequence-specific DNA binding (3:11), glucose transmembrane transporter activity (4:4), and substrate-specific transmembrane transporter activity (3:5). The Biological Process class included 862 DEGs (339:523) enriched to 39 GO terms. In the class, top 10 terms were biological process (183:268), oxidation-reduction process (42:64), transmembrane transport (38:48), metabolic process (25:20), proteolysis (4:26), methylation (5:8), nucleoside metabolic process (2:10), fatty acid biosynthetic process (8:2), protein catabolic process (2:6), and glucose import (4:4). Aside from the mentioned terms, many small GO terms containing no more than 10 DEGs were involved in specific, but important, cellular function or process. The KEGG analysis resulted in significant enrichments of 389 DEGs (153:236) to 21 pathways (Figure 5C, Appendix A). These pathways were involved mainly in carbon metabolism (19:19), glycolysis/gluconeogenesis (10:11), galactose metabolism (1:8), pentose phosphate pathway (5:5), amino acid (tryptophan, tyrosine, lysine, glycine, serine and threonine) metabolism (36:67), fatty acid degradation (10:17), metabolism (11:5) and biosynthesis (9:3), glycerolipid metabolism (5:18), sphingolipid metabolism (4:7), pyruvate metabolism (4:12), glyoxylate and dicarboxylate metabolism (5:7), propanoate metabolism (5:7), peroxisome (10:14), and ABC transporters (5:13). Notably, the up/down ratios of all gene counts implicated that most of the KEGG pathways crucial for carbon/nitrogen metabolisms were differentially repressed in the absence of *cre1*.

The validity of the transcriptomic dataset was clarified by comparing anti-log_2_ *R* values of 22 DEGs with their transcript levels in the 3-day-old SDAY cultures of Δ*cre1* relative to the WT strain. As illustrated in Figure 5D, the up/down and neutral trends of those genes revealed by qPCR analysis were consistent with most of their anti-log_2_ *R* values despite more or less differences between the two methods used to assess transcript levels of the genes involved in hydrophobicity, catalase activity, asexual development, cell wall composition or signaling and transcriptional regulation. The comparative analyses confirmed an acceptable reliability for the *cre1*-specific transcriptome.

Aside from the GO and KEGG analyses, 626 DEGs (226:400), not including those functionally unknown, were associated with main phenotypes observed in the Δ*cre1* mutant or direct/indirect role of Cre1 in mediating cellular processes and events critical for the fungal lifecycle in vivo and in vitro (Table 1 and Appendix A). The dysregulation of 166 genes (63:103) involved in carbon and nitrogen metabolisms was obviously causative of the mutant’s severe defects in radial growth on rich and scant media and also in biomass accumulation in plate and liquid cultures. The mutant’s inability to kill the model insect via NCI or BCI was associated with the malfunction of not only these genes but also 46 others (15:31) encoding virulence factors and various chitinases, proteases and lipases likely involved in cuticle degradation. As a special example, expression of bassianolide nonribosomal peptide synthetase coding gene (BBA_02630) reported as a virulence factor of *B. bassiana* [58] was abolished (log_2_ *R* = −12.23) more thoroughly than that of deleted *cre1* (log_2_ *R* = −10.45). The increase of the mutant’s sensitivity to oxidative stress correlated with 38 dysregulated genes (12:22) encoding enzymes crucial for antioxidant activity and redox homeostasis. Interestingly, only a few downregulated genes were linked to the mutant’s severe conidiation defect, including *frq1* and *vvd* required for the fungal conidiation [59,60], but none of them were key CDP activators. Twenty of 31 DEGs involved in cell wall composition/integrity were upregulated, coinciding well with unaffected cell wall integrity and no cell-lytic phenotype observed in the Δ*cre1* mutant. Seven genes encoding heat shock proteins were all suppressed. Up to 158 DEGs (55:103) were involved in cellular transport and homeostasis that may affect various biological aspects. The remaining dysregulated genes encode transcription factors (23:52) and enzymes or proteins involved in posttranslational modifications and chromatin remodeling (18:31), DNA splicing, repair, reverse transcription and translation (6:13), and various cellular signaling pathways (9:22), which are all crucial for the direct/indirect mediation of global gene expression. Altogether, Cre1 played a genome-wide regulatory role essential for *B. bassiana*’s adaptation to host insect and environment.

## 4. Discussion

Our experimental data demonstrate an indispensability of Cre1 for *B. bassiana*’s NCI and proliferation in vivo by hemocoel colonization. The indispensability relied mainly upon the Cre1-mediated CCR to make use of scant integument nutrients during hyphal invasion into insect body and of hemolymph nutrients during the proliferation in vivo leading to host death from mummification. Cre1 also mediated part of the fungal antioxidant activity crucial for scavenging reactive oxygen species (ROS) generated from host immune defense during the cuticle infection and hemocoel colonization [52]. However, the CDP activator genes essential for aerial conidiation were not evidenced as direct targets of Cre1 despite the occurrence of severe conidiation defect in the absence of its coding gene. The comprehensive effect of Cre1 on the fungal insect-pathogenic lifestyle is discussed below.

First of all, our Δ*cre1* mutant lost a whole ability to infect the model insect via NCI in repeated bioassays, highlighting an essentiality of Cre1 for the fungal NCI-dependent insect pathogenicity. This extremely compromised phenotype is different from a loss of partial virulence caused by the deletion of *cre1* from another *B. bassiana* strain [37] or *M. robertsii* [38]. The abolished NCI was linked in part to the reduced secretion of cuticle- degrading enzymes but not associated with any change in conidial hydrophobicity and adherence to insect cuticle. The reduced secretion was evidenced with repressed expression of various cuticle-degrading enzyme genes twice more of those upregulated (derepressed). The unaffected conidial hydrophobicity/adherence was evidenced with an intact outermost rodlet-bundle layer of conidial coat and differential expression of five *hyd* genes, of which *hyd1* and *hyd2* required for hydrophobin biosynthesis and assembly into rodlet bundles [39] were not downregulated in both qPCR and transcriptomic analyses. This scenario appears to differ from fungal insect pathogenicity reduced (not abolished) by disruption of the SET1/KMT2-cored pathway elucidated previously. In the studies, deletion of *set1* led to abolished or nearly abolished expression of *cre1*, *hyd1* and *hyd2*, reduced conidial hydrophobicity/adherence and attenuated virulence in *B. bassiana* [40]; deletion of *kmt2* resulted in sharp repression of *cre1* and *hyd4* required for appressorial formation and pathogenesis of *M. robertsii* [38]. The upregulation of *hyd1* and *hyd2* and no change of conidial hydrophobicity/adherence in our Δ*cre1* mutant suggest that the hydrophobin family genes required for conidial adherence and NCI could be regulated by alternative pathways not necessarily comprising Cre1.

Moreover, the abolished CBI implicates an inability for the Δ*cre1* mutant to colonize insect hemocoel by yeast-like budding for acceleration of host death. Previously, conidia injected into insect hemocoel were observed to be encapsulated by aggregated hemocytes during the first 48 h of germination and growth, followed by the release of hyphal bodies for rapid proliferation in vivo [61]. The breaking of hemocytic encapsulation is a process of fungal cells’ collapsing host immune defense by scavenging ROS derived from the defense [52]. In the fungus-insect interaction, SODs enable decomposition of superoxide radical anions into oxygen and H_2_O_2_, which is further decomposed into oxygen and water by catalases and/or peroxidases. In the present study, marked repression of critical catalase genes (*cat2* and *cat5*) concurred with decreased catalase activity, implying that increased H_2_O_2_ accumulation could impede the first-step reaction and increase an accumulation of superoxide anions. As a consequence, the Δ*cre1* mutant exhibited markedly increased sensitivities to menadione and H_2_O_2_ during colony growth, and its conidial germination was prolonged with increasing concentrations of either oxidant. In the bioassays, abundant hyphal bodies released by the control strains and lysing hemocytes were observed in the hemolymph samples at 72 h after injection, contrasting with no hyphal body released by Δ*cre1* and many intact host hemocytes consistently aggregated even at 288 h after injection. Therefore, the mutant’s hypersensitivity to oxidative stress was causative of its inability to initiate hemocoel colonization via CBI or after entry into host hemocoel via NCI. This inference is evidenced with the repressed peroxisome pathway and the low up/down ratio of many DEGs involved in cellular antioxidant response and redox homeostasis.

Furthermore, Cre1-mediated CCR is critical for fungal utilization of nutrients in insect integument and hemolymph. This is revealed from the Δ*cre1* mutant by both severe growth (reduced by 53–82%) defects on all rich/minimal media tested and largely reduced biomass accumulation in plate (SDAY) and liquid (CDB-BSA) cultures. So widely severe growth defects are close (not identical) to some of those shown by the previous Δ*BbcreA* [37] or *Magnaporthe*
*oryzae* Δ*creA* mutant [62], but very different from an essentiality of *cre1* for the cell viability of *F. oxysporum* [28], no impact of deleted *creA* on the growth of *A. brassicicola* [27], facilitation of hyphal growth and branching by *mig1* mutation in *P. funiculosum* [29], and formation of stouter hyphae in the absence of *creA* in *H. insolens* [30]. Our observations are also different from the reduced (27–42%) growth of the *T. reesei* Δ*cre1* mutant only on five of 95 carbon sources and its facilitated (31–58%) growth on nine carbon sources including galactose and maltose [24], both of which suppressed greatly the Δ*cre1* mutant’s growth in this study. However, our Δ*cre1* mutant displayed no sign of the main cell-lytic phenotype observed in the Δ*BbcreA* mutant, which was slightly responsive to osmotic stress and cell wall perturbation but not examined for antioxidant response [37]. Importantly, our transcriptomic data unveil much more comprehensive role for Cre1 in transcriptional regulation of various gene sets in *B. bassiana* than what has ever been revealed in other fungi. For example, 250 genes were shown to be differentially expressed in the *T. reesei* Δ*cre1* mutant grown in the presence of glucose, including many transporter and functionally unknown genes [24]. In fission yeast, transcriptomic analysis of Δ*scr1* (*scr1*^−^) versus wild type grown under glucose- present conditions revealed a very high up/down ratio (70:11) of 81 dysregulated genes involved in carbon metabolism, hexose uptake, gluconeogenesis and tricarboxylic acid cycle [8]. All DEGs identified from the *M. oryzae* Δ*creA* mutant compromised severely in conidiation, appressorial formation and virulence also had a very high up/down ratio of 1506:576 [62]. Interestingly, transcriptomic analysis of 852 dysregulated genes (402:450) in *Aspergillus fumigatus* Δ*creA* unveil that CreA-mediated CCR is essential for the fungal metabolic plasticity and fitness to dynamic infection microenvironment but not required for pulmonary infection establishment [63]. In contrast with the previous transcriptomic datasets, the majority of 1881 DEGs were downregulated in our Δ*cre1* mutant, leading to differential repression (denoted by up/down ratios less or much less than 1) of most enriched GO terms and KEGG pathways crucial for carbon/nitrogen metabolisms, cellular transport and homeostasis, and direct/indirect gene regulation. The previous and present studies demonstrate differential genome-wide effects of Cre1/CreA orthologues on fungal adaptation to specific lifestyle and environment. In *B. bassiana*, Cre1 is required for the utilization of scant nutrients in insect integument during NCI and of trehalose as a main carbon source of insect hemolymph during hemocoel colonization. Taken together with the Δ*cre1* mutant’s sensitivity to oxidative stress, it is not surprising to see its inability to kill the model insect via NCI or CBI.

## 5. Conclusions

Cre1-mediated CCR is essential for *B. bassiana*’s utilization of various insect-sourced nutrients and decomposition of superoxide anions and H_2_O_2_ generated from host immune defense during cuticle degradation and hemocoel colonization. The severe growth defects of our Δ*cre1* mutant on all tested media are a result of the repressed multiple pathways required for carbon/nitrogen metabolisms and cellular transport and homeostasis. It is the sharply reduced biomass accumulation in the mutant’s cultures that caused the marked reduction of conidial yield and the reduced secretion of various enzymes involved in cuticledegradation. These findings provide a novel insight into profound effect of Cre1 on the fungal insect-pathogenic lifestyle and also unravel a scenario distinctive from those of Cre1/CreA orthologues elucidated previously in other fungi.

## Figures and Tables

**Figure 1 jof-07-00895-f001:**
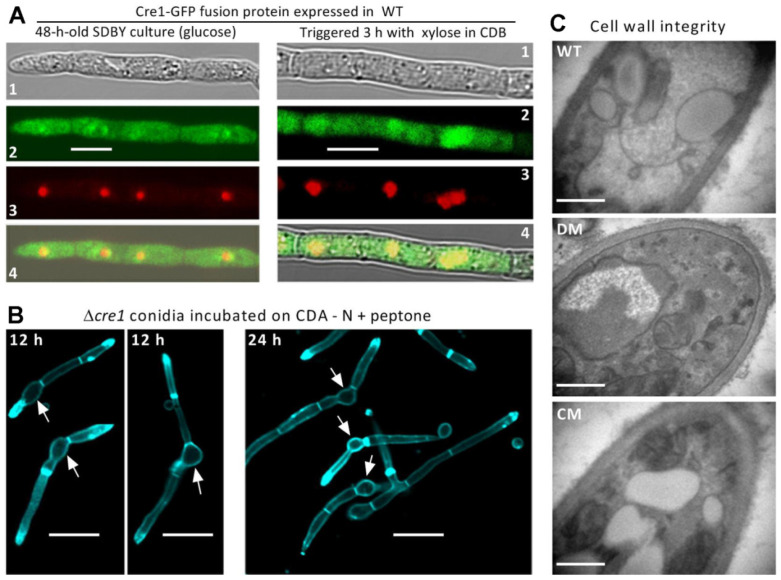
Subcellular localization of Cre1 and its role in the cell lysis of *B. bassiana*. (**A**) LSCM images (scale bar: 5 μm) for subcellular localization of Cre1-GFP fusion protein in the hyphal cells stained with the nuclear dye DAPI (shown in red) after collection from a 48-h-old culture grown in SDBY containing glucose (left panels) and then triggered for 3 h with xylose (right panels). Panels 1, 2, 3 and 4 are bright, expressed, stained and merged views of the same field. (**B**) LSCM images (scale bar: 10 μm) of germ tubes formed after 12 or 24 h incubation of Δ*cre1* conidia (arrowed) on CDA amended with peptone as sole nitrogen source. Note no sign of cell lysis occurring at the base of germ tube. (**C**) Cell wall integrity presented by TEM images (scale bar: 0.5 μm) of ultrathin sections from the hyphal cells of the wild-type (WT), Δ*cre1* (DM) and Δ*cre1::cre1* (CM) strains.

**Figure 2 jof-07-00895-f002:**
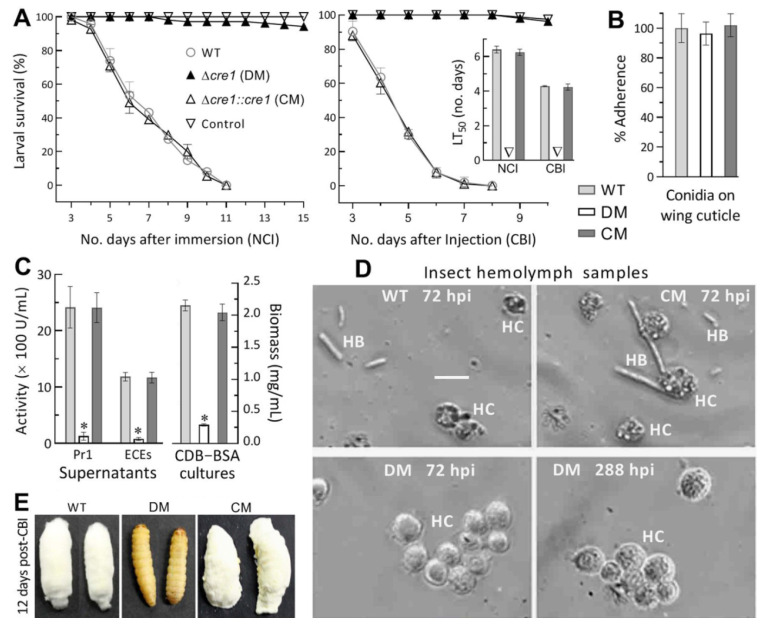
Role of *cre1* in normal infection and related cellular events of *B. bassiana*. (**A**) Time-survival trends of *G.*
*mellonella* larvae after topical application (immersion) of a 10^7^ conidia/mL suspension for normal cuticle infection (NCI) and intrahemocoel injection of ~500 conidia per larva for cuticle-bypassing infection (CBI) and LT_50_s (no. days) estimated from the trends. (**B**) Conidial adherence to locust hind wing cuticle assessed as percent ratios of pre-wash counts over post-wash counts with respect to the WT standard. (**C**) Total activities of cuticle-degrading ECEs and Pr1 proteases quantified from the supernatants of 3-day-old CDB-BSA cultures and biomass levels measured from the same cultures initiated by shaking 10^6^ conidia/mL suspensions at 25 °C. (**D**) Microscopic images (scale: 20 μm) for the presence and abundance of hyphal bodies (HB) and host hemocytes (HC) in hemolymph samples taken from surviving larvae 72 and 288 h post-injection (hpi). Note that the Δ*cre1* mutant (DM) produced no hyphal body at all even at 288 hpi, accompanied by consistent aggregation of host hemocytes. (**E**) Images of hyphal outgrowths formed by control strains (WT and CM) on the surfaces of insect cadavers and of living status of larvae 12 days pest-CBI with DM. * *p* < 0.001 (**C**). Error bars: SDs of the means from three independent replicates.

**Figure 3 jof-07-00895-f003:**
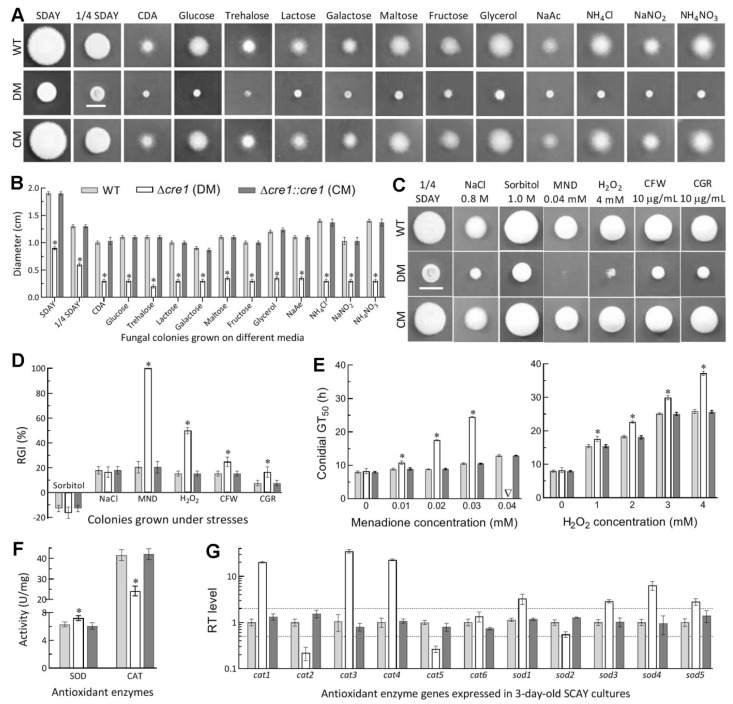
Impacts of *cre1* deletion on radial growth under normal and stressful conditions. (**A**,**B**) Images (scale: 10 mm) and diameters of 6-day-old colonies grown at 25 °C on rich SDAY, 1/4 SDAY, minimal CDA and CDAs amended with different carbon or nitrogen sources. (**C**,**D**) Images of fungal colonies grown on 1/4 SDAY alone (control) or supplemented with the indicated concentrations of chemical stressors (MND, menadione; CGR, Congo red; CFW, calcofluor white) and relative growth inhibition (RGI) percentages of fungal strains under the stresses. All colonies were initiated by spotting 1 μL aliquots of a 10^6^ conidia/mL suspension. (**E**) GT_50_ estimates for 50% of conidial germination at 25 °C on TPA alone (control) or supplemented with 0.01–0.04 mM of menadione and 1–4 mM of H_2_O_2_ respectively. (**F**) Total activities of superoxide dismutases (SOD) and catalases (CAT) quantified from the protein extracts of 3-day-old SDAY cultures. (**G**) Relative transcript (RT) levels of SOD and CAT genes in the 3-day-old SDAY cultures of *cre1* mutants with respect to the WT standard. The dashed lines indicates a significant level of one-fold down- or upregulation. * *p* < 0.05 in Tukey’s HSD tests. Error bars: SDs of the means from three independent replicates.

**Figure 4 jof-07-00895-f004:**
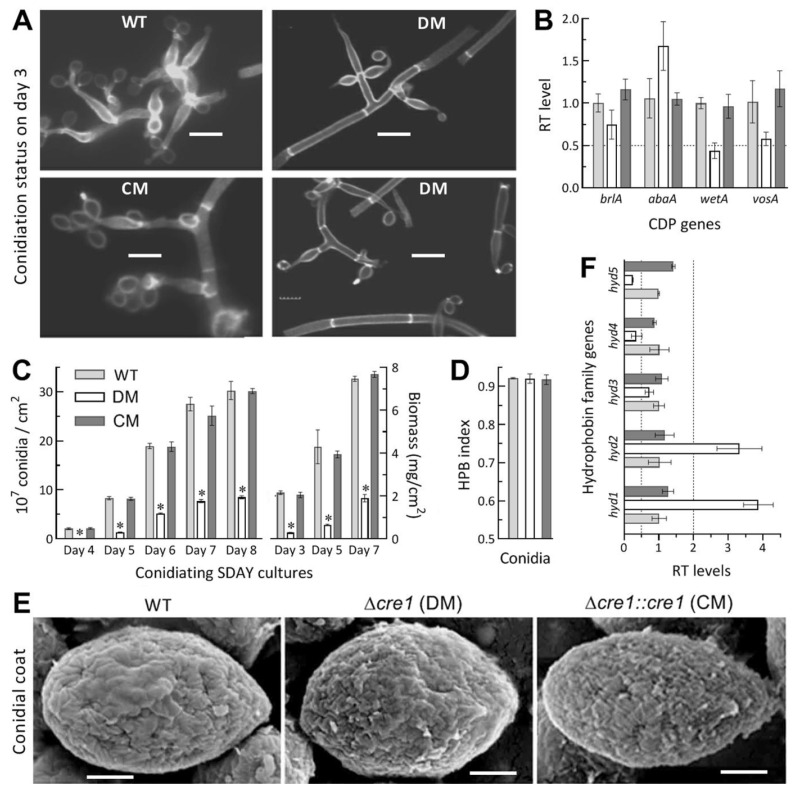
Role of *cre1* in sustaining aerial conidiation and conidial hydrophobicity of *B. bassiana*. (**A**) Microscopic images (scale: 5 μm) for conidiation status of samples taken from 3-day-old SDAY cultures and stained with the cell wall specific dye calcofluor white. (**B**) Relative transcript (RT) levels of three CDP genes and downstream *vosA* in the 3-day-old SDAY cultures of *c**re1* mutants with respect to the WT standard. (**C**) Conidial yields and biomass levels quantified from the SDAY cultures during an 8-day incubation at the optimal regime of 25 °C and L:D 12:12. (**D**) Conidial hydrophobicity (HPB) index quantified in an aqueous-organic system. (**E**) SEM images (scale: 0.5 μm) for microstructures of outermost hydrophobin rodlet bundles on conidial surfaces. (**F**) RT levels of hydrophobin family genes in the 3-day-old SDAY cultures of *cre1* mutants with respect to the WT standard. All SDAY cultures were initiated by spreading 100 μL of a 10^7^ conidia/mL suspension per plate. The dashed line (**B**,**F**) indicates a significant level of one-fold down- or upregulation. * *p* < 0.001 in Tukey’s HSD tests. Error bars: SDs of the means from three independent replicates.

**Figure 5 jof-07-00895-f005:**
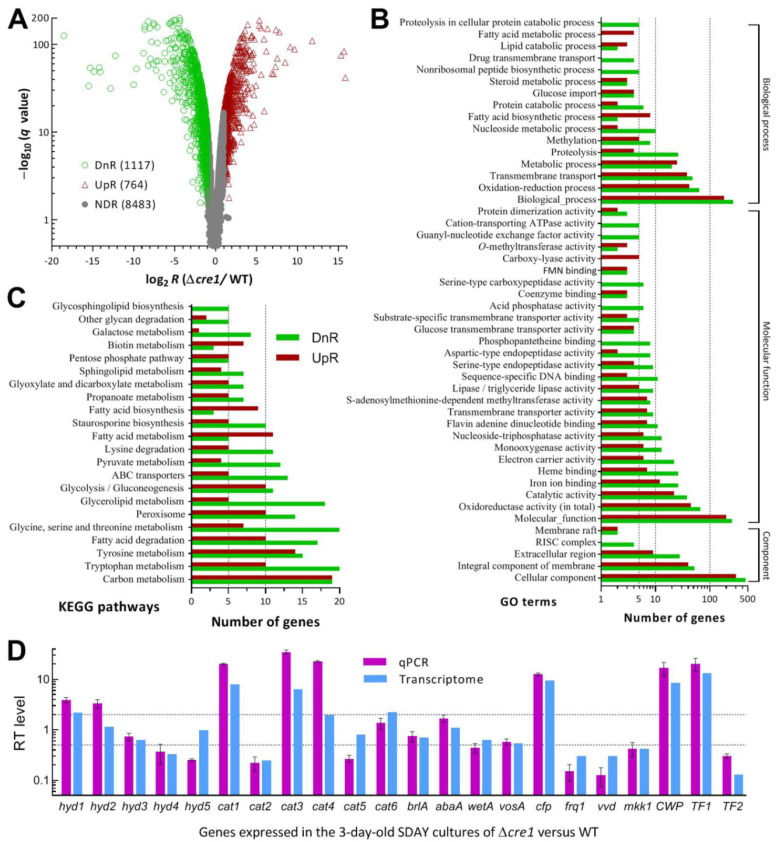
Genome-wide regulatory role of Cre1 in *B. bassiana*. (**A**) Distributions of log_2_ *R* (Δ*cre1*/WT) values versus −log_10_ *q* values for all genes significantly upregulated (UpR; log_2_ *R* ≥ 1, *q* < 0.05), downregulated (DnR; log_2_
*R* ≤ −1, *q* < 0.05) or not differentially regulated (NDR; log_2_ *R* < 1 or > −1, *q* ≥ 0.05 if log_2_ *R* > 1 or < −1) in the transcriptomes generated from three 3-day-old SDAY cultures (replicates) of Δ*cre1* and WT grown at the optimal regime of 25 °C and L:D 12:12. (**B**,**C**) Counts of differentially expressed genes (DEGs) significantly enriched to GO function classes (only main GO terms shown) and KEGG pathways, respectively. (**D**) Comparison of anti-log_2_ *R* values of 22 DEGs with their transcript (RT) levels in the 3-day-old SDAY cultures of the Δ*cre1* mutant relative to the WT standard. The upper and lower dashed lines denote the significant levels of log_2_ *R* = 1 and log_2_ *R* = −1, respectively. Error bars: SDs of the means from three cDNA samples analyzed via qPCR with paired primers (Appendix A).

**Table 1 jof-07-00895-t001:** Genome-wide impact of *cre1* deletion on cellular processes and events crucial for the adaptation of *B. bassiana* to host insect and environment.

Cellular Processes and Events	Number of Genes		
	Down	Up	Total
Carbon/nitrogen metabolisms and energy conversion	103	63	166
Host infection- and virulence-related events	31	15	46
Asexual development (aerial conidiation)	5	1	6
Response to oxidative stress	22	16	38
Cell wall composition/integrity	11	20	31
Thermal tolerance (response to heat)	7	0	7
Cellular transport and homeostasis	103	55	158
Transcriptional regulation (transcription factors)	52	23	75
Postranslational modifications and chromatin remodeling	31	18	49
DNA splicing, repair, reverse transcription and translation	13	6	19
Cellular signaling	22	9	31

## Data Availability

All data presented in this study are included in the paper and associated Appendix A. All transcriptomic data aside from those reported in Appendix A of this paper are available at the NCBI’s Gene Expression Omnibus under the accession GSE184973 (https://www.ncbi.nlm.nih.gov/geo/query/acc.cgi?acc=GSE184973, accessed on 19 October 2021).

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
