# Peer review of "Genome-Wide Insight into Profound Effect of Carbon Catabolite Repressor (Cre1) on the Insect-Pathogenic Lifecycle of Beauveriabassiana"

_jof, 2021, doi:10.3390/jof7110895_

Round 1

Reviewer 1 Report

This is a well-written MS containing interesting information. I have no suggestion for improvement.

In this study, the authors performed a functional analysis of Cre1 and reported that Cre1 mutation significantly reduces the expression of genes important for cellular homeostasis, such as antioxidant enzymes to response reactive oxygen species(ROS).

It is beyond my expertise to assess whether the phenotype of Cre1 mutant fungi are properly addressed, they appear to be, and this part is both well presented and easy to follow.

Overall, the research well planed and is logically described. I recommend this manuscript for publication in JoF.

Author Response

Please see attached a file for authors' response.

Reviewer 2 Report

The manuscript is well written, goals and hypothesis are clearly defined, results are analyzed and interpreted correctly and discussion is adequate. However, authors should choose one way how they abbreviate carbon catabolite repressor A (CreA/Cre1) and use this abbreviation through the whole manuscript. For example, Cre1 was used in the abstract, and below in the introduction, CreA was mostly used. It is not completely clear is there any reason to switch from one term to another.

Specific remarks

Introduction. In this part, it should be better indicated the difference between the current study and the study provided by Luo et al., 2014. What questions are still left unanswered by previous work?

Results. It is interesting that results of Δcre1 mutant phenotypic analysis in the current study and ΔBbrei1 mutant in Shao et al., 2019 have many similar characters (defects in radial growth, hypersensitivity to oxidative stress, abolished pathogenicity). It would be perfect to hear views on this. Do these methods fit assigned tasks?

Manuscript

minor revisions

Line 14 (here and below, plus cognate words). Hemocoel looks better for me

Line 29. change relies to rely

Line 15-16. change pathogenicidty to pathogenicity

Line 138. strain ATCC90517 to ATCC 90517

Line 176. strain ATCC90517 to ATCC 90517

Line 251 change to diphasic

Line 273. This line provides us a reference to previous work [Shao et al., 2019] about how RNA was prepared for transcriptomic analysis. However, below authors gave a more detailed description of RNA preparation for qPCR, but it also presents in [Shao et al., 2019]. Maybe these parts should be harmonized.

Line 278. No need for the abbreviation “FPKM” as it used once in the text

Line 465. Change one of two “hyd1” to “hyd2”

Line 597. Downregulated

Line 629. Mediated

Supplementary material

Page 1

  1. “cre1mutants” write separately
  2. Abbreviation “DEGs” needs to be defined here.

Author Response

(The authors gave the same response as above.)
